# Can Nitric Oxide-Based Therapy Be Improved for the Treatment of Cancers? A Perspective

**DOI:** 10.3390/ijms241713611

**Published:** 2023-09-02

**Authors:** Birandra K. Sinha

**Affiliations:** Mechanistic Toxicology Branch, Division of Translational Toxicology, National Institute of Environmental Health Sciences, NIH, Research Triangle Park, NC 27709, USA; sinha1@niehs.nih.gov; Tel.: +1-984-287-3382

**Keywords:** nitric oxide, cancer, drug delivery, liposomes

## Abstract

Since the early observations that nitric oxide (^•^NO) at high concentrations is cytotoxic to cancer cells and that it may play an important role in the treatment of human cancers, a significant number of compounds (NO-donors) have been prepared to deliver ^•^NO to tumors. ^•^NO also sensitizes various clinically active anticancer drugs and has been shown to induce the reversal of multi-drug resistance in tumor cells expressing ATP-binding cassette-transporter proteins. For the successful treatment of cancers, ^•^NO needs to be delivered precisely to tumors, and its adverse toxicity must be limited. Like other chemotherapeutics, the precise delivery of drugs has been a problem and various attempts have been made, such as the encapsulation of drugs in lipid polymers, to overcome this. This prospective study examines the use of various strategies for delivering ^•^NO (using NO-donors) for the treatment of cancers. Finding and utilizing such a delivery system is an important step in delivering cytotoxic concentrations of ^•^NO to tumors without adverse reactions, leading to a successful clinical outcome for patient management.

## 1. Introduction

The growth and survival of cancer cells in vivo is complex and involves many pathways. Tumors, in general, are heterogeneous, containing many different types of cells, including cancer stem cells, which are resistant to therapy. Furthermore, cancer cells develop resistance during therapy. Therefore, successful cancer treatment in the clinic has been difficult. Nitric oxide (^•^NO) has emerged as a promising therapeutic for the treatment of many human cancers [1,2,3]. ^•^NO, a free radical produced endogenously by nitric oxide synthase (NOS), has a relatively short half-life (2.0–0.18 s) under physiological conditions [4]; thus, it is difficult to deliver it selectively to cellular targets. As a free radical, ^•^NO chemically reacts with cellular protein-SH and many other free radicals in biological environments, producing adverse reactions [5]. Additionally, different concentrations of ^•^NO have different physiological effects: at a low concentration, ^•^NO can be used to regulate vasodilation, nerve transmission, anti-inflammatory responses, and antioxidant responses, and it also promotes tumor growth [6,7,8]. In contrast, at high concentrations, ^•^NO inhibits tumor growth and enhances the apoptosis/ferroptosis of tumor cells via the formation of both reactive oxygen and reactive nitrogen species [8]. Because of this double-edged nature of ^•^NO in the regulation of physiological processes, ^•^NO must be delivered selectively to its target sites to limit off-target adverse effects.

Various molecular structures and chemical materials have been developed to exogenously deliver ^•^NO using synthetic NO donors (e.g., S-nitrosothiols (RSNOs), N-diazeniumdiolates, and metal–NO complexes); a number of reviews are available [9,10,11,12,13,14,15,16]. The breakdown and release of ^•^NO are dependent upon the molecular structure of the donor, which allows the rate of release and effective targeting of ^•^NO to be controlled in biological systems. This breakdown property of NO-donors has been utilized therapeutically by exploiting the microenvironment of cancers, as they exhibit lower pH values due to altered cellular metabolisms. Thousands of NO-donors have now been synthesized and tested in various tumor models. Many of these compounds show significant activity in terms of inhibiting tumor cell growth in both in vitro and in vivo tumor models. A number of tumor-specific NO-donors, e.g., J-SK (*O*^2^-(2,4-dinitrophenyl) 1-[(4-ethoxycarbonyl)piperazin-1-yl] diazen-1-ium-1,2-diolate), which are activated only in tumors and are effective in killing tumor cells without causing significant toxicity, have also been synthesized [17,18]. 

^•^NO has been shown to sensitize drug-resistance tumor cells to chemotherapeutic drugs, e.g., adriamycin and cis-platin [19,20]. This would indicate that highly active NO- donors (including encapsulated) can be utilized in combination with other therapeutic modalities, including chemotherapy, radiation therapy, photothermal therapy, and photodynamic therapy. Studies from our laboratory have also shown the utility of ^•^NO generating compounds in reversing resistance mediated by the overexpression of ATP-binding cassettes.

(ABC)-transporters [21,22]: Research has shown that the overexpression of a family of efflux proteins is responsible for chemotherapy resistance [23,24]. These efflux proteins remove drugs from the cancer cells in an energy-dependent manner and belong to the ABC-transporter superfamily. We have shown that ^•^NO directly inhibits the functions of ATPase activities of both topoisomerase II [25] and ABC-transporters [26] following the nitrosylation of cysteines. This inhibition of ATPase functions in multi-drug-resistant tumor cells results in the enhanced cellular accumulation of drugs and the reversal of resistance [26]. Our findings are important, as they suggest that ^•^NO-based therapy can be utilized to overcome drug-resistance, which is one of the major challenges facing the treatment of cancers in clinics. 

Some ^•^NO-generating compounds, such as nitroglycerine (GTN) and nitrates, are currently utilized in the clinic for the treatment of acute angina [27,28,29]. GTN has also been shown to exhibit a positive response in combination with vinorelbine and cis-platin [30]. Furthermore, a slow-release transdermal GTN patch has also been shown to be effective in men with prostate tumors whose disease was progressing following surgery or radiotherapy [31]. In addition, dietary arginine (which increases ^•^NO formation) has been utilized in combination with radiation therapy in some clinical trials [32]. However, while thousands of NO-donors have been synthesized and tested as anti-cancer drugs, and some have demonstrated significant anti-cancer activities in vivo, very few of these newer compounds have been evaluated in the clinic, and none of the NO-donors are currently utilized as anticancer drugs (or in combination) in the clinic. 

Because high concentrations of ^•^NO are required for effective tumor cell killing, the need to deliver ^•^NO precisely to tumor targets is of paramount importance. Here, I examine the use of various delivery systems for NO-donors that may improve the selective delivery of cytotoxic concentrations of ^•^NO for the treatment of cancers. The major goal of this perspective is to examine the current and future delivery systems of NO-donors (^•^NO) to tumors that may lead to therapeutic effects. It appears that the synthesis of additional NO-donors alone will not advance the treatment of cancers, as improvements in delivery systems are urgently needed for the future of ^•^NO-based therapy (as a single agent or in combination) in the clinic.

## 2. Prodrug Strategy for NO-Donors

Strategies using prodrugs have been effective in solving many drug delivery problems. In order to develop prodrugs, it is extremely important to have a balance between stability and the ability to rapidly convert prodrugs into their active forms under normal physiological conditions. It is believed that, in healthy tissue where pH homeostasis is maintained near pH 7.4, cancer tissues will promote a rapid ^•^NO release at the site due to lower pH (4–6), thereby mitigating off-target cytotoxicity. Various NO-donors (prodrugs) that are activated by cellular enzymes to release ^•^NO at tumor sites have also been designed and synthesized. These NO-releasing prodrugs are activated by various cellular enzymes, including esterase and glutathione transferase (GST) [18,33]. These NO prodrugs have been shown to have a significant advantage over direct non-specific NO-donors, as they reduce their side effects. Of importance for the treatment of cancers are NO-prodrugs that are activated by glutathione-dependent GST, as a number of tumors, including drug-resistant tumor cells, overexpress these enzymes. 

Huang et al. [9] synthesized a number of O2-(Sulfonylethyl-derived) diazeniumdiolates which were active against GST-expressing tumor cells. Some of these compounds were shown to be significantly better than JS-K, an originally synthesized GST-activated NO-donor prodrug. We have also utilized JS-K in our studies and found this compound to be highly active against P-gp- and GST-expressing multi-drug-resistant ovarian tumors [21]. In addition, we found JS-K to be effective in reversing P-glycoprotein 170 (P-gp)-mediated drug resistance against multi-drug-resistant (MDR) tumor cells. Furthermore, we found esterase-mediated activation of the prodrug NCX4040 to be extremely cytotoxic to both P-gp-expressing ovarian and breast cancer resistance protein (BCRP)-expressing breast tumor cells. It also caused the reversal of both P-gp- and BCRP-mediated drug resistance in cells [22]. NCX4040, originally synthesized as an anti-carcinogenic compound, was found to be active against human colon cancer cells and did not display any significant toxicity in vivo. 

It should be noted that these NO prodrugs have neither been fully evaluated in clinical trials nor are currently being utilized in the clinic for the treatment of cancers.

## 3. Antibody–Drug Conjugates for Targeted Delivery of ^•^NO in Tumors

Monoclonal antibody drugs with high target recognition and affinity have emerged as important treatment modalities for cancers. However, they lack sufficient therapeutic efficacy and have a narrow therapeutic window. Therefore, the strategy involving antibody/peptide–drug conjugates (ADC/PDC) has been utilized recently [34]. An antibody–drug conjugate (ADC) is a targeted drug delivery system composed of an antibody linked to an active cytotoxic drug via a lysine or -SH function. Since the antibody specifically recognizes tumor cells, it can deliver the drugs directly to the surfaces of tumor cells, thereby enhancing the anticancer effect and increasing the therapeutic efficacy. It is important that the target antigen be highly or preferentially expressed on the cancer cell surface (and minimally expressed in healthy tissue) to bind with the circulating ADC. It is also important that the target antigen have minimal secretion in circulation from the cancer cells, as antibodies can bind to these secreted receptors in the circulation, reducing the amount of antibodies available for target cell binding and internalization. Hence, selecting an appropriate target antigen is a critical step for the success of ADCs.

Recently, Sun et al. [35] prepared a conjugated humanized cluster of differentiation 24 (CD24) monoclonal antibody with a NO-donor which was connected to the antibody using an -SH function. Conjugates between the antibody and the NO-donor were stable in the blood circulatory system, with a low concentration of GSH. However, it was readily cleaved in tumor cells containing a high amount of GSH to generate two molecules of ^•^NO. The NO–antibody conjugate showed significant cytotoxicity, both in vitro and in vivo, against hepatic carcinoma cells, with fewer side effects on normal tissues. 

More studies are needed to expand this concept of antibody–NO conjugates in order to provide a clinically active drug conjugate. It will also be useful to prepare antibody–drug conjugates of other highly active NO-delivery compounds, e.g., JS-K analogs and NCX4040, and to examine their effectiveness against a variety of tumor models. 

Ruthenium–NO complexes have also been utilized for the targeted delivery of ^•^NO to tumors, resulting in the delivery of high concentrations of ^•^NO in tumor cells [36]. Studies using *cis*-[RuII(NO+)Cl(dcbpy)2]2 conjugated to a polyclonal antibody showed enhanced cytotoxicity against the liver hepatocellular carcinoma cell line HepG2. These metal complexes were stable in tumor cells and delivered cytotoxic concentrations of ^•^NO. In another study, [RuII(bpy)2L(NO+)]n+ was utilized as an NO-donor against the mitochondria of cancer cells [37]. Mitochondria are extremely important, as they provide cellular energy and play an important role in cell survival and death. The voltage-dependent anion channel (VDAC) plays a central role in mitochondria-mediated apoptosis and ferroptosis by generating reactive oxygen species (ROS). The formation of ROS is significantly increased in the presence of ^•^NO. Therefore, VDAC has been utilized as a potential target for ruthenium complex NO-donors, and [RuII(NO+)Cl(dcbpy)2]3−anti-VDAC antibody conjugates have been prepared and evaluated as anticancer drugs. The cytotoxicity of the *cis*-[RuII(NO+)Cl(dcbpy)2]3−-anti-VDAC antibody was significantly enhanced in comparison to the free *cis*-[RuII(NO+)Cl(dcbpy)2]3−complex. This cytotoxic effect was found to result from a site-specific interaction of the complex with VDAC and the release of ^•^NO. 

While Ru–NO complexes may potentially be useful as anticancer drugs, the accumulation of NO–ruthenium complexes in tumor cells is a problem due to the lack of solubility at pH values higher than 4.0. This needs to be addressed before these complexes can be utilized in clinical settings. 

## 4. Liposomes-Based Delivery of NO-Donors 

The encapsulation of therapeutics within liposomes is an effective strategy for the controlled delivery of drugs to their targets. Liposomes are nanostructures composed of an inner aqueous core surrounded by a bilayer of phospholipids, and have been used to enhance drug delivery against many diseases, including cancer [38]. Liposomes, as drug delivery vehicles, provide many benefits, e.g., reduced immune response, increased cellular uptake, and protection of drug payload from premature actions or breakdown. Clinically active drugs, e.g., adriamycin, paclitaxel, and cis-platin, have serious side effects that limit their full potential in the clinic. For example, adriamycin causes cumulative dose-dependent cardiotoxicity that can be life-threatening, and is, therefore, dose-limiting [39]. Similarly, paclitaxel and cis-platin show peripheral neuropathy [40] and nephrotoxicity [41], respectively, in humans. As potent and effective cytotoxic drugs, these chemotherapeutic agents have benefited notably from encapsulation into nanoparticles, which significantly improves their specificity toward cancer cells, concomitantly decreasing their toxicity and thus allowing for the administration of higher doses to patients. 

The preparation of NO-releasing liposomes utilizing alkaline interior aqueous phases enables the encapsulation of varying concentrations of NO with long shelf-lives (>3 months). The use of liposomal NO donors as anticancer agents has shown increased therapeutic potential over small-molecule NO-donors [42,43]. NO-releasing liposomes represent an important strategy for cancer treatment due to their pH-triggered release. It has been suggested that varying the lipid bilayer content (e.g., phospholipid composition, charge) may further enhance the stability of this NO-delivery system and influence NO-release payloads. In addition, utilizing a more thermally stable lipid (e.g., 1,2-distearoyl-*sn*-glycero-3-phosphocholine) could potentially prolong the NO-release duration even further due to its greater bilayer rigidity. Earlier works have focused on utilizing S-nitrosothiol or N-diazeniumdiolate NO-donors in liposomal delivery systems [44]. It would be very interesting to encapsulate recently developed, highly active analogs of anticancer NO-donors in FDA-approved liposomes. Such liposomal NO-donors may provide a much better opportunity for the successful treatment of cancers in the clinic.

Recent studies employing multilayered liposomes have shown increased stability of the core drug due to its unique fabrication methodology [45]. In this technique, the liposomal layers are layered above each other using a layer-by-layer technique, resembling onion-like layering. It is expected that encapsulated NO-donors will have enhanced protection from decay and will have increased sustained release to specific tumor targets. However, at this time, a multilayer liposome delivery system has not been tried with NO-donors. Thus, there is significant need to utilize this multi-layering approach to encapsulate newer and more active NO-releasing compounds for evaluation as improved anticancer drugs. 

## 5. Exosomes as a Delivery System for NO-Donors

The use of exosomes as drug delivery systems is a recent and emerging field [46,47,48]. Exosomes are bilayer membrane-coated extracellular vesicles measuring between 40 and 100 nm in diameter, and are considered natural nanoparticles which can carry a cargo of a wide variety of compounds, e.g., drugs, lipids, proteins, RNAs, and DNAs. Exosomes have been proposed for drug-delivery purposes, as they can be loaded with both small molecules and macromolecules, which can be used as therapeutic tools to treat cancers. Exosomes play an important role in intercellular communications, and as natural carriers, exosomes have the advantages of low immunogenicity and direct delivery of drugs to cells. Exosomes also have many desirable features, such as long circulating half-lives, the intrinsic ability to target tissues, biocompatibility, and minimal or no inherent toxicity, overcoming the limitations observed with the majority of other delivery systems. Recent studies have shown the simplicity of isolation and the possibility of commercial scale-up and utilization for tumor-targeting. These features make exosomes ideal nanoparticles for drug delivery, with wide therapeutic applications. In addition, exosome-based delivery systems have been shown to bypass the effects of ABC-transporter proteins and to overcome the multi-drug resistance of tumor cells [49]. 

The key to using exosomes as drug carriers, however, is the effective loading of exogenous drugs (chemotherapeutics and NO-donors) into exosomes. Currently, sonication, electroporation, transfection, incubation, extrusion, saponin-assisted loading, transgenesis, freeze–thaw cycles, thermal shock, pH gradient methods, and hypotonic dialysis have been applied to load drugs into exosomes [50]. The clinical application of exosomes as drug carriers (and NO-donors) depends upon methods of importing drugs into exosomes efficiently. Furthermore, selecting the correct source of exosomes is very important, as certain cancer-cell-derived exosomes have been shown to have pro-cancer and pro-resistance effects, making them unsuitable as drug carriers [51].

At the moment, the delivery of NO-donors using exosomes has not been attempted or reported for the treatment of diseases, including cancers. We have recently initiated an exosome system for delivery with selective highly active NO-donors; however, it too early for any conclusions to be made. 

## 6. Conclusions and Future Directions

Nanomedicine faces many challenges, especially in clinical trials. In clinical applications, ^•^NO, as a gas messenger that can act on the entire body, requires careful consideration due to its physical and chemical properties and its accumulation in non-target organs and tissues, which pose considerable toxicity. Furthermore, some NO nanomedicines may cause oxidative stress, immune responses, protein oxidation, and DNA damage. The existing NO-donors use the targeting properties of materials to release ^•^ NO at a specific site, and certain NO-donors can be used for slow release over time, but they still lack the precise control needed to release appropriate concentrations of ^•^NO. As ^•^NO is a double-edged sword, proper concentrations of ^•^NO must be utilized to treat tumors; otherwise, ^•^NO could promote further aggravation of the disease or other toxic manifestations. Therefore, it is necessary to design and develop smarter NO nanomedicines that can release the appropriate concentrations of ^•^NO in vivo. It has been suggested that we need to design smart NO-donors that can be turned on or off to release appropriate amounts of ^•^NO according to the pathophysiological requirements. In this scenario, when they are turned on, they continuously release the desired concentration of ^•^NO at the tumor site, and when they are turned off, the release of ^•^NO stops. Furthermore, it is necessary that the NO nanomedicine remains localized within the tumor until the next time the smart NO-donors are turned on without breakdown. Although the characteristics of nanomaterials can be used to achieve the coordinated release of ^•^NO, various strategies have also been developed for the design of NO-releasing materials, including the incorporation of photo-activable, charge-switchable, or bacteria-targeting groups [52,53]. While these strategies have led to the development of some excellent NO-donors and nanomaterials, none of these have found utility in the treatment of cancers. 

Because ^•^NO is different from most other anticancer drugs and it has some diverse effects, the delivery of specific concentrations of ^•^NO to tumors is essential to mitigate the adverse effects. However, unlike other chemotherapeutics, it is cytotoxic to all kind of tumors, and can sensitize various clinically active anticancer drugs. In addition, it can also inhibit the ATPase functions of ABC transporters [22,25], resulting in the reversal of multi-drug resistance. These properties of NO-generating compounds make them extremely desirable as anticancer drugs. Therefore, a significant number of NO-donors have been synthesized for the strict purpose of curing human cancers; many of these compounds are active in killing tumor cells in vitro at sub-micromolecular ranges and also show significant activity in vivo mouse tumor models. What is most interesting is that several nitric oxide-releasing drugs have been utilized for a long time to treat cardiovascular diseases, including isosorbide mononitrate, which is a long-term nitric oxide-releasing drug administered to patients with hypertension [28]. Furthermore, other nitric oxide-releasing drugs, e.g., nitroglycerin and sodium nitroprusside (SNP), are also currently used as short-acting anti-angina drugs to rapidly decrease blood pressure [27]. While these drugs are used in millions of patients every day, no nitric oxide-releasing drugs are currently utilized for treatment of cancers in the clinic.

Advances in drug delivery system(s) do offer exciting prospects in terms of packaging active NO-donors into the clinic and beyond. Significant improvements have been made in the field of encapsulation of chemotherapeutics [54,55]. The FDA has now approved various liposomes for therapy which should be tried with highly active anticancer NO-donors. Our recent studies indicate that certain NO-donors induce ferroptosis-mediated tumor cell death and that combinations of erastin or RSL3 (Ras-selected ligand3) with NO-donors enhances tumor cell killing [56]. Erastin and RSL3 are small molecules known to induce ferroptosis in tumor cells. These studies further suggest the utility of NO-donors and ^•^NO in killing tumor cells. Hence, the utilization of encapsulated NO-donors in combination with ferroptosis inducers may be of significance for the treatment of Ras-mutated tumors in the clinic. While approaches utilizing antibody–drug conjugate delivery systems are expensive and difficult in terms of obtaining a high NO-payload, these methods are highly specific for the precise delivery of cytotoxic concentrations of ^•^NO to tumors. It is my opinion that ADC may be the future of NO-donor (drug) delivery, and may be able to mitigate off-target toxicity. Furthermore, recent advances in the study of exosomes for the delivery of chemotherapeutics also suggest that NO-donors can be packaged into exosomes for the treatment of human diseases, including cancers. Figure 1 summarizes the possible roles of these strategies in the future of NO nanomedicine for the treatment of human cancers.

Finally, there is also a significant need to test NO-donors and NO-medicine in reliable models to predict the outcomes in patients. It is difficult and unethical to test and compare nanomedicines against free drugs in patients. Currently, much of the cytotoxicity testing is conducted in 2D-tissue culture systems, and is sometimes combined with humanized mouse models. Traditional 2D cell culture systems do not mimic the 3D culture systems, as they lack the complexity of the tumor microenvironments in patients and often fail to predicate patient outcomes [57,58,59,60]. Developing 3D model systems to test NO-donors and NO-nanomedicines would be a step forward for more accurately predicting the therapeutic outcomes. It would also be advantageous to develop 3D model systems from patient-derived tumor tissues and stem cells for predicting the outcomes of NO nanomedicine (and NO-donors) for both personalized medicine and for overcoming drug resistance in the clinic.

## Figures and Tables

**Figure 1 ijms-24-13611-f001:**
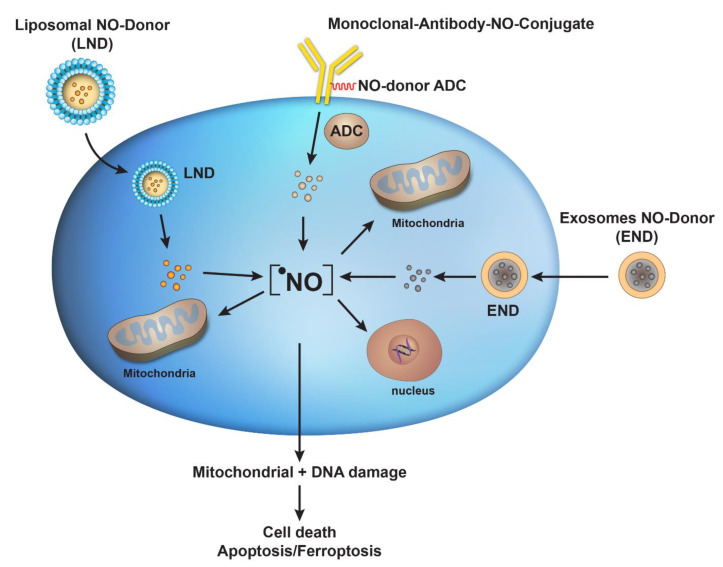
Various delivery systems for NO nanomedicine utilizing liposomal-encapsulated NO-donors (LNDs), antibody-NO-drug conjugates (ADCs), and exosome-encapsulated NO-donors (END) for the treatment of cancers in the clinic. Shown here is a tumor cell in which these nanomedicines containing NO-donors are taken up by passive transport (endocytosis) and release their NO-donor payloads. NO-donors then release cytotoxic •NO, which combines with ROS (O2−) to generate peroxynitrite. Both ROS and RNS induce damage to tumor mitochondria and nuclear DNA, causing cellular death.

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
