# Peer review of "Can Nitric Oxide-Based Therapy Be Improved for the Treatment of Cancers? A Perspective"

_ijms, 2023, doi:10.3390/ijms241713611_

Round 1

Reviewer 1 Report

The use of nitric oxide for anticancer therapy seems quite promising and worthy of attention. The main problem is associated with high potential toxicity and non-targeted effects of the active substance, so its encapsulation is a logical way out. The author reviews possible delivery vehicles, including liposomes, exosomes and molecular constructs. As emphasized by the author, in addition to encapsulation, it is necessary to be able to release the encapsulated nitric oxide at the desired location, i.e. implement release on demand.  However, progress in this area remains unclear. The part of the article that could be improved is this area. It is not clear whether such solutions exist, whether the encapsulation with controller release is implemented for nitric oxide. I can suggest to add more information and references covering this area. Are there already working approaches for controlled release and how do they work? It's really critical.

Author Response

The reviewer is correct in suggesting that an encapsulated NO-donor with controlled release (on command) would make an ideal candidate for delivering specific amounts of NO to targets and for the treatment of human cancers. However, this has not been reported and therefore, was not discussed in this perspective.

Reviewer 2 Report

In this manuscript, the author discusses nitric oxide (NO)-based treatment of cancers in particular focusing on NO-releasing biomolecules to aid in NO delivery. Regarding NO-related cancer treatments, there has been so far many reports on their therapeutic effects as well as their side effects. Although I don't think it's a review with particularly new contents, it is a concise manuscript focusing on the recent methods for delivering the exact amount of NO to cancer cells that are precisely the target, and raising problems to overcome as well.

Minor points

There are some paragraphs for the references to be added in order to better supplement their contents.

Page 2 line 72-73, “In addition, dietary arginine…..in some clinical trials.”

Page 3 line 117-118, “Therefore, the strategy involving…. (ADC/PDC) has been recently utilized.”

Page 4 line 169-172, “For example, adriamycin …. and nephrotoxicity, respectively in humans.”

Page 5 line 218-221, “Currently, sonication……have been applied to load drugs into exosomes.”

Page 6 line 248-252, “Although the characteristics…  have found utility in treatment of cancers.”

Page 6 line 256-267, “In addition, it can also inhibit ATPase …..to rapidly decrease blood pressure.” which is necessary to list up the literature corresponding to each clause.

In addition to these listed above, I think there are other clauses in the text that need to be referenced.

Author Response

More references have now been added.